# Emerging Role of Interferon-Induced Noncoding RNA in Innate Antiviral Immunity

**DOI:** 10.3390/v14122607

**Published:** 2022-11-23

**Authors:** Jie Min, Wenjun Liu, Jing Li

**Affiliations:** 1CAS Key Laboratory of Pathogenic Microbiology and Immunology, Institute of Microbiology, Chinese Academy of Sciences, Beijing 100101, China; 2Savaid Medical School, University of Chinese Academy of Sciences, Beijing 100049, China; 3Institute of Microbiology, Center for Biosafety Mega-Science, Chinese Academy of Sciences, Beijing 100101, China

**Keywords:** noncoding RNA, interferon, interferon-stimulated gene, antiviral innate immunity

## Abstract

Thousands of unique noncoding RNAs (ncRNAs) exist within the genomes of higher eukaryotes. Upon virus infection, the host generates interferons (IFNs), which initiate the expression of hundreds of interferon-stimulated genes (ISGs) through IFN receptors on the cell surface, establishing a barrier as the host’s antiviral innate immunity. With the development of novel RNA-sequencing technology, many IFN-induced ncRNAs have been identified, and increasing attention has been given to their functions as regulators involved in the antiviral innate immune response. IFN-induced ncRNAs regulate the expression of viral proteins, IFNs, and ISGs, as well as host genes that are critical for viral replication, cytokine and chemokine production, and signaling pathway activation. This review summarizes the complex regulatory role of IFN-induced ncRNAs in antiviral innate immunity from the above aspects, aiming to improve understanding of ncRNAs and provide reference for the basic research of antiviral innate immunity.

Following Crick’s central dogma [1,2], genetic information is passed from DNA to RNA, which is then translated into proteins. Over many years, proteins have been widely recognized as the final products that perform the main functions encoded by genes, though protein-coding genes occupy <2% of the human genome. As sequencing technology advances, other products of the human genome are coming into view, revealing the “dark energy” of DNA. According to the NIH ENCODE project results, approximately 80% of the human genome is endowed with biochemical functions and, interestingly, about 62% of genes are transcribed into noncoding RNAs (ncRNAs) [3]. Upon viral infection or interferon (IFN) stimulation, many ncRNAs are generated to regulate various vital cellular processes, including pre-transcriptional, transcriptional, and post-transcriptional regulatory processes. Numerous ncRNAs are reported to be involved in antiviral activities, most of which are microRNAs (miRNAs), long ncRNAs (lncRNAs), or circular RNAs (circRNAs).

IFN was originally discovered as an infection-inhibiting factor. Subsequently, IFN-specific inducible mRNAs and proteins were identified. The canonical interpretation of IFN-mediated antiviral innate immune responses suggests that IFNs induce the transcription of IFN-stimulated genes (ISGs), which further alter the proteome in cells and establish an antiviral cellular state [4]. However, according to recent studies, IFN-mediated antiviral innate immune responses may be a more complex process than transcriptional induction of ISGs. Initially, researchers mainly detected IFN-induced gene expression by surveying IFN-stimulated transcriptome changes through microarray technology, which enabled the detection of the enrichment of select mRNAs through poly(A) affinity or arrays used to probe for transcripts of encoded proteins [5,6,7]. Thus, they were unable to capture ncRNAs that were not polyadenylated, such as miRNAs, circRNAs, piwi-interacting RNAs (piRNAs), and enhancer RNAs (eRNAs, which are lncRNAs) [7,8,9,10]. Polyadenylated lncRNAs have been frequently dismissed as “garbage” because they do not encode proteins. However, an increasing number of studies indicate that IFNs can induce some ncRNAs to participate in antiviral innate immunity [11,12,13,14,15]. These results demonstrate the biogenesis of ncRNAs and the role of ncRNAs, enhancing our understanding of these molecules and serving as fundamental concepts to guide antiviral innate immunity.

## 1. The Antiviral Innate Immune Response of the Host

Innate immunity is the first and most rapidly formed line of defense against viral invasion. Host cells recognize pathogen-associated molecular patterns (PAMPs) through pattern recognition receptors (PRRs). After viral nucleic acid is recognized by host cells, a series of signaling pathways is activated, stimulating IFN expression and secretion. IFNs induce hundreds of ISGs, including antiviral proteins and ncRNAs, which activate antiviral innate immune defenses.

### 1.1. The Pattern of Host Recognition of the Virus

According to their location in a cell, PRRs are divided into two classes. One is composed of a family of toll-like receptors (TLRs) located on the endosomal membrane. A major member of the TLR family, TLR3, recognizes double-stranded RNA, while TLR7 and TLR8 recognize single-stranded RNA, and TLR9 recognizes nonmethylated CpG DNA [16]. The other class consists of three nucleic acid receptors that are widely expressed in the cytoplasm. They include AIM2-like receptors (ALRs) and cyclic GMP-AMP synthase (cGAS), which recognize DNA viruses, and retinoic acid-like receptors (RLRs), which recognize double-stranded RNA or 5′ppp-modified single-stranded RNA [17,18,19].

### 1.2. IFN Signaling Pathways

IFNs are members of the class II cytokine family and are classified into three categories based on their molecular structure, antigenicity, and cell source. Type I IFNs are mainly composed of IFN-α produced by leukocytes and fibroblast-generated IFN-β. IFN-ε, IFN-ω, and IFN-κ also belong to this family. Type III IFNs include IFN-λ1, IFN-λ2, IFN-λ3, and IFN-λ4, which have antiviral mechanisms similar to those of type I IFNs [20,21,22]. The type II IFN family, also known as the immune IFN family, has only one member, IFN-γ, which is mainly produced by natural killer (NK) cells, T lymphocytes, and antigen-presenting cells (APCs) [23].

As shown in Figure 1, type I IFNs elicit signals through IFN-α/β receptor subunit 1/2 (IFNAR1/2), and different subtypes exhibit inconsistent tissue expression and binding affinity for IFNAR1/2 receptors, leading to various biological reactions [24]. Type III IFNs bind IL-10R2 of the interleukin-10 receptor and type III IFN-λ receptor 1 (IFNLR1) to activate the JAK-STAT signaling [25,26]. Type I IFN receptors are extensively distributed on the surface of nucleated cells. The type III IFN receptor complex IL-10R2 is ubiquitous, but the IFNLR1 subunit is only expressed in specific cells and tissues. Research has revealed that IFN-III receptors are also distributed in greater quantities in epithelial cells than in other cells and are involved in mucosal immunity [27]. In addition, with either autocrine or paracrine function, three different types of IFNs bind to high-affinity receptors (IFNAR2, IFNGR1, and IFNLR1) on the surface of the cell, recruiting low-affinity receptor subunits (IFNAR1, IFNGR2, and IL-10R2) to produce complexes that can transmit signals, further inducing the expression of hundreds of antiviral proteins encoded by ISGs and the transcription of antiviral effector ncRNAs, thereby activating antiviral innate immune defenses [28].

Several groups simultaneously found that a variety of ncRNAs are induced by type I, II, and III IFNs [29,30,31,32]. For example, STAT3 homodimers indirectly suppress proinflammatory gene expression by facilitating suppressors of cytokine signaling 3 (SOCS3) expression [33,34] and miR221/222 expression [35,36,37], or induce hitherto unknown transcriptional repressors. These studies strongly suggest that there is a close relationship between IFN-mediated antiviral innate immunity and ncRNAs.

### 1.3. Features and Functions of ISGs

Approximately 2000 ISGs in humans and mice have been identified to date. However, most of these have not been functionally characterized [38]. Analyses of >300 known ISGs indicate that they mainly play an immunomodulatory role. They can directly induce antiviral activity, especially by enhancing host cell recognition of and response to pathogens, and can negatively regulate IFN signaling pathway activity. To successfully infect a host, a virus needs to enter a target cell to replicate, assemble, and proliferate. ISGs produced by host cells are directly involved in the antiviral response. These ISGs target the key steps of virus infection and inhibit virus replication.

In addition, most PRRs that recognize viruses and the molecules involved in IFN signaling are ISGs, such as protein kinase R (PKR), cGAS, and the IFN regulator 1/3/7/9 (IRF1/3/7/9). ISGs are expressed at a basal level when not stimulated by exogenous factors. However, once a pathogen is recognized, downstream signaling pathways are activated and the expression of many ISGs is induced, including molecules involved in other PRRs and IFN signaling pathways. If not quickly eliminated or overactivated, IFNs produce signals that may lead to chronic inflammation or autoimmune diseases. Thus, host cells activate and produce a series of ISGs that negatively regulate IFN signaling to maintain host immune balance. Furthermore, post-translational modification of the virus and host proteins exerts an impact on viral replication. For example, ISG15 is one of the most highly expressed ISGs, which can inhibit herpes virus, influenza virus, and coxsackievirus B3 infection [39]. ISG15 is a 15-kDa ubiquitin-like protein that plays a role in the post-translational modification of hundreds of viral and host proteins [40,41].

## 2. The Role of NcRNAs in Antiviral Innate Immunity Regulation

Each stage of the viral life cycle is affected by the antiviral proteins encoded by ISGs. According to the life cycle of the virus, antiviral ISGs can be broadly divided into three categories: ISGs that encode proteins that prevent virus entry into cells, inhibit virus replication and translation, or prevent virus release from cells (Figure 2). Moreover, some ISGs can participate in antiviral innate immunity through their transcribed ncRNAs [42,43]. These ncRNAs usually regulate transcription, splicing, and nucleic acid degradation by acting as a decoy, and they modulate translation via RNA–DNA, RNA–RNA, or RNA–protein interactions. IFN-induced ncRNAs can also be used as novel antiviral effectors to participate in antiviral innate immunity.

NcRNAs are roughly sorted into short ncRNAs (fewer than 200 nt) and lncRNAs (more than 200 nt). A number of miRNAs and lncRNAs have been reported to be involved in antiviral activity. The miRNAs constitute a class of short ncRNAs of ~22 nt in length that participate in different biological processes through the post-transcriptional regulation of genes. The lncRNAs are grouped into intronic lncRNAs, long intergenic lncRNAs (lincRNAs), antisense lncRNAs, pseudogene lncRNAs, and eRNAs based on their position in the genome or relationship to mRNA. In addition, emerging circRNAs are classified as lncRNAs that are generated by back splicing and 3′ to 5′ end self-ligation. The lncRNAs can act as miRNA sponges, scaffold for macromolecules, decoy proteins, and transcription regulators. Notably, most ncRNAs do not encode proteins, but encode small functional polypeptides. Interestingly, we note that a class of ncRNAs regulated by IFN may be involved in antiviral innate immunity. Therefore, it is meaningful to classify IFN-induced ncRNAs as ISG complement products into interferon-stimulated non-coding RNA (ISR).

### 2.1. Effect of IFN-Induced miRNAs

The effect of miRNA on mRNA translation and stability is well established. Previous studies have attempted to explore the role of miRNAs in viral infection, and found that the interaction between miRNA and the RNA viral genome directly regulates viral pathogenesis [44]. To date, two outcomes of these interactions have been identified, both of which directly affect virus replication. One is to prevent viral replication by inhibiting viral translation; the other is to maintain the stability of viral RNA to enhance replication. Moreover, miRNA-mediated changes in protein expression alter the host’s response to infection.

#### 2.1.1. IFN-Induced MiRNAs Directly Affect Virus Replication

Furthermore, IFN-induced miRNAs directly target viral transcripts. IFN-β-regulated miRNAs substantially reduce hepatitis C virus (HCV) replication and are sufficient to impose an antiviral cellular state. A customized microarray containing 245 human and mouse miRNAs was used to characterize IFN-regulated miRNAs in Huh7 cells, and the expression of 30 miRNAs was found to be either induced or repressed. The seed sequences of eight up-regulated miRNAs (miR1, miR30, miR128, miR196, miR296, miR351, miR431, and miR448) were complementary to those in HCV RNA, which was a surprising result. In addition, IFN inhibits miR122 expression and thus positively controls HCV replication [13]. Notably, miR-122 is critical for maximum HCV replication, and strategies based on blocking this miRNA to prevent HCV replication have shown promise in both in vivo models and preliminary clinical trials [45]. Furthermore, miR122 can target hepatitis B transcripts [46], whereas miR29 targets human immunodeficiency virus (HIV) transcripts [47]. Both miR122 and miR29 were previously identified as IFN-regulated miRNAs.

#### 2.1.2. IFN-Induced MiRNAs Mediate the Expression of Host Proteins

Altering miRNA transcription or activity may lead to important consequences for IFN responses, and recent reports suggest that IFN stimulation can affect cellular miRNA expression [15]. For example, IFN-γ-activated STAT1 induces the expression of miR-155 via directly binding to the miR-155 promoter [48]. STAT1 binds to the miR-146a promoter, leading to the inhibition of miR-146a expression [49]. In addition, miR-155 expression induced by viral infection targets SOCS1 (a STAT inhibitor) to inhibit SOCS1 activity and generates feedback to promote IFN-I-mediated antiviral activity [48,50].

MiRNAs may indirectly down-regulate the expression of ISGs, thereby antagonizing the IFN-I response. For example, miR-221 expression, caused by up-regulation of miR-221 expression in peritoneal macrophages induced by viral infection, negatively regulates the innate antiviral response to vesicular stomatitis virus (VSV) [35]. In addition, STAT3 stimulates miR-221/222 expression, which targets PDLIM2 to stabilize and increase STAT3 levels [36]. Further, inhibition of miR-221/222 up-regulates the expression of components of the IFN-α signaling pathway (including STAT1, STAT2, IRF9, and several ISGs) in human glioma cell lines [37]. Hence, miR-221/222 can indirectly regulate ISG expression by affecting the balance of STAT1/STAT2 verses STAT3 homodimer signaling.

### 2.2. Effect of IFN-Induced LncRNAs

The regulatory effect of IFN-induced lncRNAs on antiviral proteins has been confirmed. In response to viral infection or IFN stimulation, many lncRNAs are produced, which regulate various activities as shown in Figure 2 and Table 1. The roles of these IFN-induced lncRNAs in regulating ISG *cis*-transcription, ISG *trans*-expression, ISG translation, and IFN expression are reviewed below.

#### 2.2.1. IFN-Induced LncRNAs Regulate ISGs *cis*-Transcription

IFN-induced lncRNAs can be co-expressed with adjacent ISGs to regulate the latter in *cis*. Some lncRNAs originate from bidirectional promoters shared with ISGs or are located nearby (<2 kb) immune-related protein-coding genes [60]. For example, Tetherin is a transmembrane protein encoded by the ISG bone marrow stromal antigen 2 (*BST2*), and through the unique topology of this protein, HIV-1 viral particles can be trapped for subsequent degradation in endosomes or lysosomes, preventing viral budding [61]. ISG-encoded BST2 can also inhibit the release of progeny influenza viruses [62]. The expression of lnc*BST2* depends on the JAK-STAT signaling pathway. The lncRNAs *BST2* and BST2 share the same promoter and are co-expressed upon IFN stimulation or influenza infection to promote the expression of the antiviral protein ISG BST2/tetherin [31,53].

In contrast, lncRNA *IFI6* negatively regulates IFI6 promoter function by modifying histones to inhibit IFI6 expression, thereby increasing HCV replication [11]. The lncRNA *IFI6* is an *IFI6-*transcribed lncRNA with expression induced by IFN-α or HCV. However, the lncRNA *IFI6* does not exert its regulatory effect through the JAK-STAT signaling pathway.

The loci of IFN- and influenza A virus (IAV)-inducible (PR8ΔNS1), ISR2, ISR8, and lncISG15 are close to those of ISGs GBP1, IRF1, and ISG15, respectively [31,32]. After IAV infection, the expression levels of ISR2, ISR8, and lncISG15 are significantly increased, and the expression levels of downstream genes display the same up-regulated trend [31,32]. In addition, the expression of ISR12 is strongly induced by IFN at late times, ISR12 is located upstream of IL-6, and the ISR12 promoter has a binding site for NF-κB [32]. These studies reveal an important role for ISR2, ISR8, ISR12 and ISG15 in viral infection and IFN regulation. However, the underlying mechanisms remain uncertain.

#### 2.2.2. IFN-Induced LncRNAs Regulate ISGs *trans*-Expression

IFN-induced lncRNAs also can regulate ISGs in *trans*. For example, lncRNA #32 leverages the interaction between hnRNPU and ATF2 to regulate ISG expression and positively regulate the host antiviral response. Interestingly, the lncRNA is negatively regulated by IFN-β [55]. IFN-α2 or the IAV can induce lncRNA EGOT activity. The EGOT lncRNA negatively regulates antiviral response by inhibiting the expression of ISGs (GBP1, ISG15, and Mx1) [54].

In addition, IFN-induced lncRNAs differentially affect the expression of ISGs in different types of cells. The lncRNA NRIR is also known as the lncRNA *CMPK2*. LncRNA *CMPK2* activity is induced by IFN or HCV in hepatocytes, and it is a negative regulatory factor of ISG-induced inhibition of HCV replication [29]. Hantavirus (HTNV) infects epithelial cells in the same manner that HCV infects hepatocytes, and NRIR is a negative regulator of infection promoters. For example, IFITM3 inhibits HTNV infection, but IFITM3 action is negatively regulated by NRIR [51]. Thus, lncRNA CMPK2 has high cell- and induction stimulus-specificity. The expression of repressed ISGs in hepatocytes or epithelial cells is re-established by up-regulation lncRNA *CMPK2* induced in monocytes to promote lipopolysaccharide (LPS)- induced activation of IFN signaling [52]. Together, these results confirm the role of lncRNA *CMPK2*/NRIR as a negative regulator (hepatocytes and epithelial cells) and a positive regulator (monocytes) of the IFN response.

#### 2.2.3. IFN-Induced LncRNAs Regulate ISGs Translation

As competitive endogenous RNAs (ceRNAs), lncRNAs can trap miRNAs. Some lncRNAs contain miRNA-targeted mRNA sequences, and lncRNAs contain miRNA-targeted mRNA sequences that are similar to miRNA-targeted mRNA sequences, which can promote the release of mRNA from miRNA and thus antagonize miRNA function. Our study shows that IFN-β- or IAV-induced lncRNA ISG20 is a ceRNA. As a common miRNA of the ISG20 gene and lncRNA ISG20, miR-326 acts on the 3′ untranslated region (UTR) of ISG20 mRNA to suppress ISG20 translation. We found that lnc-ISG20 binds to miR-326 and acts as a ceRNA, which ablates the inhibitory effect of miR-326 on ISG20 mRNA, increases ISG20 protein levels, and inhibits influenza virus replication [42].

#### 2.2.4. IFN-Induced lncRNAs Regulate IFN Expression

IFN-induced lncRNAs exert regulatory effects by modulating the transcription of target genomic loci in *trans* to inhibit IFN signaling pathway activation. Our previous study reports that ISG *MxA* transcribes a lncRNA after IAV infection or IFN-β stimulation [43]. MxA acts as a broad-spectrum antiviral by trapping viral components early during infection and preventing the virus from reaching host cells [63,64,65,66]. We found that a lncRNA–DNA triplex was formed by lnc-MXA binding to the IFN-β promoter, interfering with IRF3 and p65 binding to the IFN-β promoter, inhibiting the transcription of IFN-β in *trans*, negatively regulating RIG-I-mediated antiviral immune response, and promoting IAV proliferation [43]. IFN-induced lncRNA participates in antiviral innate immunity by inhibiting RIG-I activation. lncRNA Lsm3b binds RIG-I monomers to compete with viral RNA, and the feedback generated at the late stage of the innate reaction leads to RIG-I-induced inactivation of innate immune function. Mechanistically, lncRNA Lsm3b binding limits conformational changes in the RIG-I protein and prevents downstream signaling, terminating type I IFN production [58].

In addition, lncRNAs can be cofactors to mediate interactions between macromolecules, thereby inhibiting viral replication. Binding of lncITPRIP-1 to MDA5 promotes MDA5 oligomerization and enhances the association between MDA5 and HCV RNA, resulting in downstream MAVS that promotes signaling and produces IFN that inhibits HCV replication [56]. LncRNA Lrrc55-AS binds to phosphatase methylesterase-1 (PME-1) and promotes the interaction between PME-1 and phosphatase PP2A (an IRF3 signaling inhibitor). LncLrrc55-AS supports PME-1-mediated demethylation and PP2A inactivation, which enhances IRF3 phosphorylation and signaling pathways, promoting IFN-I production and inhibiting viral replication [57].

Moreover, there are many IFN-induced lncRNAs whose functions remain unknown. For instance, the effect of lncRNA ISR is dependent on the effect of IAV or IFN-α-induced RIG-I, and lncRNA ISR inhibits IAV replication, but the specific mechanism remains unclear [12]. Further, seven candidate IAV- and IFN-up-regulated lncRNAs have been identified: AC015849.2, RP-1-7H24.1, PSMB8-AS1, CTD-2639E6.9, PSOR1C3, AC007283.5, and RP11-670E13.5. Among them, inhibition of PSMB8-AS1 reduces IAV replication and proliferation [59].

### 2.3. Effect of CircRNAs

CircRNA is a newly discovered class of single-stranded ncRNAs that can inhibit the activity of other RNAs or RNA-binding proteins. With the continuous development of sequencing technology, many circRNAs have been identified in animals, plants, and humans. Notably, circRNAs display specific expression patterns dependent on cell type, tissue type, and developmental stage, revealing their significant regulatory role in gene expression. More importantly, circRNAs sponge miRNA, an important regulator of gene expression. For example, it is reported that circ-Vav3 expression is up-regulated in chicken hepatomas infected by avian leukemia virus J-subunit, causing sponging of gga-miR-375 to eliminate the effect of the miRNA on its target gene, YAP1, increasing YAP1 expression, and inducing the epithelial-mesenchymal transition [67]. Additionally, the alternative splicing of mRNA before it forms a circRNA affects protein production. When ectopically expressed in Kaposi’s sarcoma herpes virus (KSHV) infected cells, the circRNA hsa_circ_0001400, which is induced by KSHV infection, regulates viral gene expression without affecting viral genome replication [68]. CircRNAs can also be translated into small peptides. Carcinogenic human papillomavirus (HPV)-induced circRNAs contain part of the E7 oncogene, which is modified by N6-methyladenosine (m6A), preferentially localized in the cytoplasm, related to polysomes, and ultimately translated into the E7 oncoprotein to facilitate cancer cell growth [69]. CircRNAs impair autophagy and participate in viral infection. After infection with H1N1-subtyped IAV, the expression of circGATAD2A is up-regulated, but overexpression of circGATAD2A impairs autophagy and promotes IAV replication [70]. CircRNA produced in vitro activates RIG-I-mediated innate immune responses and confers protection against viral infection [71].

CircRNAs have antiviral effects similar to ISGs. Because circRNAs are derived from pre-mRNA, they are transcribed ISGs, and many ISG exon- or intron-specific circRNAs have been identified [72]. CircRNAs also regulate IFN signaling to participate in antiviral innate immunity. A novel intron splicing circRNA, AIVR, is up-regulated by IAV infection, which acts as a miR-330-3p sponge to release CREBBP mRNA [73]. Thus, increased cellular expression of CREBBP can accelerate IFN-β production. However, whether IFN can induce AIVR in IAV-infected A549 cells needs further study. In addition, IAV-PR8 induces circRNA_0082633 by JAK-STAT signaling activation, enhancing ISRE promoter activity to up-regulate IFNB1 mRNA levels, promoting type I IFN signaling [74]. Importantly, hundreds of circRNAs may be involved in M1 macrophage activation after IFN-γ stimulation [75].

## 3. Discussion

### 3.1. Validation of IFN-Induced NcRNAs Complements the Traditional Definition of ISGs

IFNs trigger the transcription of ISGs, change the protein composition of host cells, and mediate the antiviral state. Although IFN-mediated antiviral innate immunity has been studied for many years, more research is needed to understand IFN-regulated antiviral mechanisms. Many recent discoveries in infectomics based on contemporary sequencing and proteomic techniques have changed our understanding of IFN-related biology [4,30,76]. In a narrow sense, ISG expression is up-regulated at the transcriptional level in response to stimulation with IFN, a direct target of the IFN signaling pathway. ISGs have traditionally been thought to function through the proteins they encode, but recent studies have shown that they can also exert antiviral functions through the ncRNAs they transcribe [13,29,30,42,43]. Peng et al. [77] detected that widespread differentially expressed (DE) lncRNAs respond to virus infection and regulate the host’s innate immune response by sequencing analysis. Interestingly, most up-regulated DE lncRNAs after viral infection are also significantly up-regulated by IFN-α treatment [30,77]. Therefore, promoter analysis and expression correlation studies suggest that these lncRNAs might be ISGs [30]. For example, a recent study used ChIPBase v2.0 to reveal that the expression of lncRNAs in SARS-CoV-2-infected cells might regulate STAT1, STAT3, and IFN regulatory factors [78]. These studies have updated our understanding of the biological landscape of IFNs and lncRNAs.

Andrew et al. investigated the presence of IFN-induced ncRNAs in human cells. They found hundreds of DE ncRNAs, but by analyzing ISG-encoded transcript types, a total of 90% and 92% of up-and down-regulated genes were classified as protein-coding genes, respectively [79]. Along these lines, we propose that IFN-induced ncRNAs complement ISG products. As genome annotation becomes more complete and more ncRNAs are discovered, a deeper understanding of the role of ncRNAs in antiviral innate immunity will become possible. Of note, the expression of many ISGs is induced by IFNs, viral infection, double-stranded RNA, and signaling pathways [80]. Therefore, studying IFN-induced ISGs and ISGs triggered by pathogenic microorganisms will help to discover ISGs and reveal their antiviral mechanisms.

### 3.2. Diversity of IFN-Induced NcRNAs

The dozens of IFNs are classified into three categories. The classification of IFNs may lead to the diversity of IFN-induced ncRNAs. As seen in Figure 1, IFNs activate multiple signaling pathways through different receptors. Different signaling pathways stimulated by IFNs may induce different ncRNAs. Although the type I and type III IFN genes are distinct and bind to different receptors, they are induced by the same pathologic route of infection and activate the expression of relevant antiviral, antiproliferative, and immunoregulatory genes. Although IFN-γ is derived only from immune cells, its receptor is expressed at moderate or low levels in almost all cell types [23]. The structure of type II IFNs is independent of that of type I and III IFNs, but it induces the same transcription of some ISGs as that induced by type I IFNs [81]. Moreover, type I IFN signaling initiates IFN-γ signaling [82]. In addition, the distribution of receptors in different cells may be the reason ncRNAs exhibit specific expression patterns that depend on cell or tissue type. These results suggest that IFN-induced ncRNAs are diverse, but some of them can be induced by all types of IFNs, which may play a more important role in antiviral immunity responses.

The expression of ncRNAs largely depends on the cell type and is tightly controlled by various cellular signaling pathways. Because different IFNs induce the expression of different proteins, different IFNs specifically induce the expression of various ncRNAs. For example, numerous ISG-encoded proteins inhibit HCV replication. However, certain ISG-encoded proteins, such as ISG15 and USP18, promote HCV replication. Thus, ncRNAs act as positive or negative regulators to orchestrate a complex network and balance the IFN response, similar to the effect of ISG-encoded proteins. Furthermore, high-throughput sequencing of the transcriptome reveals that sequences in ncRNAs are similar to those reported to be in ISGs, suggesting that they originate from the transcription of the same genes. The ncRNA network of ISGs may help extend our understanding of the innate immune system against viruses. Whether IFN-stimulated ncRNAs interact with IFN-stimulated proteins to play a synergistic or antagonistic role in innate immunity remains to be further investigated.

### 3.3. Specificity of NcRNAs Induced by a Viral Infection

The characterization of ncRNAs not only provides a novel model for host defense in mammalian cells, but has also enabled the identification of novel components that can be added to the repertoire of antiviral effectors. Similar to the lncRNA *VIN* [83], these ncRNAs are induced only by specific viruses and are not affected by a general viral or IFN response, which is associated with differences in virulence. Additionally, some ncRNAs are not expressed at the same level when stimulated by different viruses, and some play opposite roles in the antiviral response. For example, lncRNA *NEAT1* is up-regulated by HIV-1 [84,85], Herpes simplex virus (HSV) [86,87], HTNV [88], hepatitis D virus [89], and IAV [86], but down-regulated by severe dengue virus [90]. In general, up-regulation of lncRNA *NEAT1* expression appears to be a general response to viral infection. However, by promoting HSV infectivity, lncRNA *NEAT1* inhibits the replication of other viruses. Whether the consequences promote or inhibit viral infection depends on downstream mechanisms.

### 3.4. Others Potential Effects of NcRNAs on Host Antiviral Innate Immunity

Many recent advances have been made in identifying ncRNA modifications and have been used to study their effects on lncRNAs and circRNAs, in addition to ribosomal RNAs (rRNAs) and tRNAs [91]. The methylation of adenosine at the N6 position in mammalian RNA is a highly abundant RNA base modification that promotes efficient initiation of translation of circular RNA proteins, participates in cellular activities, or performs other functions [92]. Therefore, we wondered whether different RNA modifications of IFN-stimulated ncRNAs could affect ncRNA function in antiviral immunity. IFN-stimulated ncRNAs are translated whether or not they participate in antiviral immunity through peptides. However, whether IFN-stimulated ncRNAs directly exert antiviral effects by targeting viral proteins remain to be determined. Both circRNAs and lncRNAs interact with miRNAs to perform various biological functions. Numerous circRNA–miRNA–mRNA interactions have been identified in viral infection, but the functions of these interaction networks remain to be further explored. Furthermore, whether IFNs induce certain other ncRNAs, such as snRNAs, rRNAs, piRNAs, and/or tRNAs, to participate in antiviral innate immunity is a worthy direction for future research. Indeed, determining whether IFNs induce these RNAs to participate in antiviral innate immunity and whether they form interaction networks warrants further investigation.

## 4. Conclusions

Given the role of IFN-stimulated ncRNAs in antiviral immunity, which enriches our understanding of IFN-mediated antiviral innate immunity, we were interested in determining the function of IFN-induced antiviral ncRNAs as ISGs. The IFN-stimulated miRNAs can directly target viral transcript or indirectly affect host protein expression to take part in antiviral innate immunity. The IFN-stimulated lncRNAs regulate gene expression on translation in *cis* or *trans*. Notably, most ncRNAs do not encode proteins, but some encode small polypeptides. We were also interested in ncRNA, which encodes peptides that add to antiviral innate immunity. Compared to other types of ncRNAs, lncRNAs exhibit surprisingly more functions. However, there are still many unanswered questions, such as whether the evolution of lncRNAs and miRNAs is conserved. In order to solve these outstanding problems, future studies need to focus on lncRNA biology and immune-related lncRNA functions in vivo. In addition, emerging circRNAs are classified as lncRNAs generated by back splicing and 3′ to 5′ end self-ligation. However, few reports address the basic mechanisms of IAV infection and replication, or the regulatory mechanisms by which circRNAs affect the interaction between IAV and host cells. Therefore, research on circRNA with respect to antiviral innate immune responses induced by influenza virus, the molecular basis of virus RNA binding proteins, the circRNA transcription regulatory mechanisms, analysis of circRNA effects on cell function, and determining the network(s) involved in signaling pathways and regulation are worthy of future research. We expect to find that circRNAs regulate cell functions and discover the mechanisms by which they influence viral replication and pathogenicity.

## Figures and Tables

**Figure 1 viruses-14-02607-f001:**
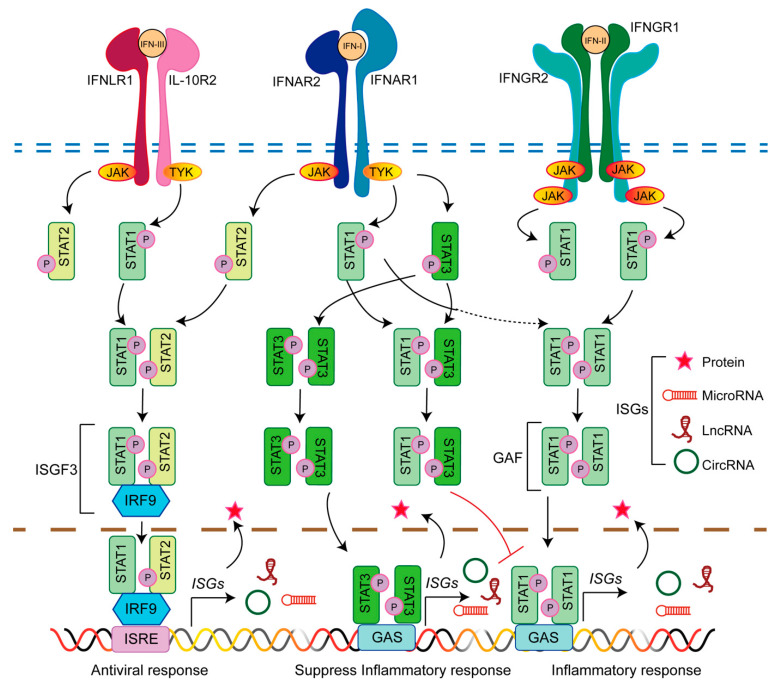
Interferon (IFN)-stimulated gene (ISG) expression. The three types of IFNs trigger similar downstream signal transduction and transcriptional reactions, albeit these reactions are mediated through different receptors. Type I IFN and type III IFN receptors actively phosphorylate JAK1 and TYK2, which phosphorylate the specific intracellular tyrosine residues of the receptors. STAT1/2/3 are recruited and phosphorylated. Type II IFN receptors trigger the phosphorylation of bound JAK1 and JAK2, and then phosphorylate receptor chains to induce the recruitment and phosphorylation of STAT1. STAT1 and 2 form a heterodimer that recruits IRF9 to form IFN-stimulated gene factor 3 (ISGF3), which is translocated into the nucleus and binds to IFN-stimulated response element (ISRE) sequences to trigger antiviral gene expression. STAT3 homodimers suppress proinflammatory gene expression. STAT1 homodimers bind to the gamma-activated sequence (GAS) to activate the inflammatory response. Phosphorylated STAT3 sequesters activated STAT1 to form heterodimers and prevent STAT1 from forming functional homodimers that trigger downstream gene expression. Gamma-activated factor (GAF).

**Figure 2 viruses-14-02607-f002:**
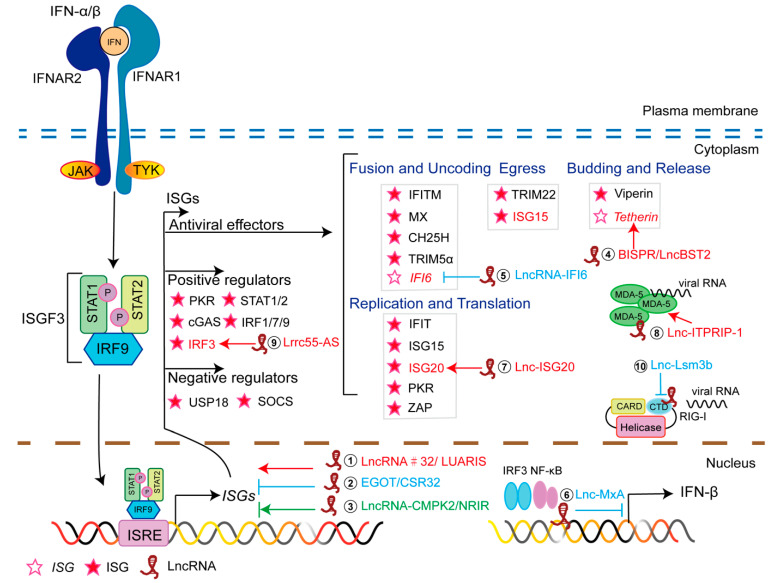
Antiviral innate immunity induced through ISGs. ISG products interfere with different stages of viral life cycles. Long noncoding RNAs (lncRNAs) act as ISGs in the regulation of antiviral innate immunity. ① lncRNA #32/LUARIS positively regulates the expression of ISGs by interacting with ATF2 and hnRNPU. ② lncRNA EGOT/CSR32 inhibits the expression of ISGs. ③ lncRNA CMPK2/NRIR suppresses ISGs transcription in hepatocytes or epithelial cells, and up-regulates the expression of ISGs in human monocytes. ④ BISPR/lncBST2 promotes the expression of the antiviral ISG BST2/tetherin. ⑤ Through its spatial domain, lncRNA IFI6 regulates histone modification at the IFI6 promoter, inhibiting the expression of IFI6. ⑥ lncRNA MxA inhibits the transcription of IFN-β. ⑦ As an endogenous competitive RNA, lnc-ISG20 binds to the microRNA (miRNA) miR-326, releasing ISG20 mRNA from the miRNA. ⑧ lncRNA ITPRIP-1 promotes the inhibitory effect of MDA5 by facilitating the binding of MDA5 to viral RNA. ⑨ lncRNA Lrrc55-AS supports PME-1-mediated demethylation and inactivation of PP2A, enhancing IRF3 phosphorylation and signaling. ⑩ lncRNA Lsm3b binds RIG-I, restricting the conformational change in RIG-I protein and thus preventing downstream signaling. Red arrow stands for promoting; blue arrow stands for inhibiting; green arrow stands for promoting or inhibiting depending on the cell type. The lncRNAs ①–⑥ are involved in antiviral factor transcription, and ⑦–⑩ affect the antiviral factors post-transcription.

**Table 1 viruses-14-02607-t001:** Interferon (IFN)-induced lncRNAs act as ISGs and are involved in the host antiviral innate immune response.

lncRNAs	Description	Stimuli	Regulation	Effect on Viral Replication	Characteristics/Functions	References
LncRNA CMPK2/NRIR	Negative regulatory factor of ISG response in hepatocytes or epithelial cells; positive regulator of the LPS-induced IFN response in human monocytes.	IFN-α, IFN-γ, HCV, LPS	Up	+/−	LncRNA CMPK2/NRIR inhibits ISG (CMPK2, CXCL10, IFIT3, IFITM1, ISG15, Viperin, and IFITM3) transcription by forming RNA–protein complexes, interacting with chromatin during remodeling, or transcription factors in hepatocytes or epithelial cells; NRIR upregulates the expression of IFN-I stimulated genes (CXCL10, MX1, IFITM3, and ISG15) in monocytes.	[29,51,52]
BISPR/lncRNA BST2	Positive regulator of ISG response.	IFN-α2, IFN-λ, IAV (PR8ΔNS1), VSV(M51R), HCV	Up	−	Promotes the expression of the antiviral ISG BST2/tetherin.	[31,53]
EGOT/CSR32	IFN signaling pathway negative regulator; induced by NF-κB after PKR or RIG-I activation.	IFN-α2, poly(I:C), IAV, HCV, SFV	Up	+	Inhibits the expression of ISGs (GBP1, ISG15, Mx1, BST2, ISG56, IFI6, and IFITM1).	[54]
LncRNA ISG20/NONHSAG017802	Positive regulator of the ISG response; has the same chromosomal location as ISG20; most of the sequences are the same.	IFN-β, IAV, SeV, poly(I:C)	Up	−	Lnc-ISG20, as an endogenous competitive RNA that binds to miR-326, releasing ISG20 mRNA, and inhibiting IAV replication.	[42]
LncRNA MxA/NONHSAG032905	Negative IFN signaling pathway regulator; in the MxA locus.	IFN-β, IAV, SeV, poly(I:C)	Up	+	Lnc-MxA negatively regulates the RIG-I-mediated antiviral immune response, inhibits the transcription of IFN-β by combining with the IFN-β promoter to form a lncRNA–DNA triplex.	[43]
LncRNA IFI6/lncRNA RP11-288L9.4	Negative regulator of the ISG response; overlaps with the antisense strand of IFI6 within intron 1 and is located in the IFI6 gene in the human genome.	IFN-α, HCV	Up	+	Through its spatial domain (large right arm), it regulates histone modification at the IFI6 promoter, inhibiting the expression of IFI6 and promoting HCV infection.	[11]
LncRNA ISR	Within the BAHCC1 locus.	IFN-β, IAV	Up	−	It relies on RIG-I signaling and inhibits IAV replication; however, the specific mechanism is still unclear.	[12]
LncRNA #32/LUARIS	Positive IFN signaling pathway regulator.	IFN-β, poly(I: C)	Down	−	It positively regulates the expression of IRF7, CCL5, CXCL11, OASL, RSAD2, and IP-10 by interacting with ATF2 and hnRNPU.	[55]
LncRNA ITPRIP-1	Cofactors of MDA5.	IFN-α, HCV, HSV, SeV, VSV	Up	−	Promotes the inhibitory effect of MDA5 on HCV replication by facilitating the binding of MDA5 to viral RNA.	[56]
LncRNA Lrrc55-AS	Positive regulator of IFN-I production.	IFN-β, poly(I:C), HSV, LPS,	Up	−	It supports PME-1-mediated demethylation and inactivation of PP2A, enhancing IRF3 phosphorylation and signaling.	[57]
LncRNA Lsm3b	Negative IFN signaling pathway regulator; multivalent structural motifs and long-stem structure.	IFN-α, IFN-β, VSV, SeV, HSV	Up	+	Its binding restricts the conformational change of the RIG-I protein and prevents downstream signaling, terminating the production of type I IFNs.	[58]
PSMB8-AS1	Near PSMB8 and TAP1.	IAV and IFN-β	Up	+	Repressed PSMB8-AS1 reduces IAV replication and proliferation.	[59]
AC015849.2	Near Chemokine (C-C Motif) Ligand 5 (CCL5) and TATA Box Binding Protein (TBP)-Associated Factor (TAF15).	IAV and IFN-β	Up	NA	NA	[59]
RP-1-7H24.1	Near OAS2, OAS3 and TRIM25.	IAV and IFN-β	Up	NA	NA	[59]
CTD-2639E6.9	An intergenic lncRNA (lincRNA).	IAV and IFN-β	Up	NA	NA	[59]
PSOR1C3	A sense intronic lncRNA that lies within introns and does not overlap with exons; near POU5F1 and HLA-C.	IAV and IFN-β	Up	NA	NA	[59]
AC007283.5	3 prime overlapping lncRNA that overlaps the 3ʹ-UTR of a protein-coding locus on the same strand; near CASP10 and CFLAR.	IAV and IFN-β	Up	NA	NA	[59]
RP11-670E13.5	Near OAS2, OAS3 and TRIM25.	IAV and IFN-β	Up	NA	NA	[59]
LncISG15	Near ISG15.	IFN-α2, IFN-λ, IAV (PR8ΔNS1), VSV (M51R), HCV	Up	NA	NA	[31]
ISR2	Located at the end of the GBP gene cluster, adjacent to GBP6, and is a pseudogene of GBP1.	IFN-α2, IFN-β, HCV, IAV (PR8ΔNS1), HIV	Up	NA	NA	[32]
ISR8	Near IRF1.	IFN-α2, IFN-β, HCV, IAV (PR8ΔNS1)	Up	NA	NA	[32]
ISR12	Near IL-16.	IFN-α2, IFN-β, TNFα, LPS, poly(I:C)	Up	NA	NA	[32]

## Data Availability

Not applicable.

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
