# Peer review of "Emerging Role of Interferon-Induced Noncoding RNA in Innate Antiviral Immunity"

_viruses, 2022, doi:10.3390/v14122607_

Round 1
Reviewer 1 Report
This manuscript reviews the roles of IFN-induced non-coding RNAs in modulating innate antiviral immunity. It appears to be comprehensive and up to date, and would be of value to investigators who are new to the field.
Comments:
1. The manuscript would benefit from some reorganization. For a review paper, the discussion is unnecessarily long and includes information that either is covered or should be covered at the beginning of the paper. I suggest that all the text discussing type I, II, and III IFNs and their respective receptors be covered at the beginning of the article. It appears to me that referring to STAT3 homodimers as suppressive requires more support than the references included in the manuscript.
2. The article would benefit from a better description at the beginning of the various types of noncoding RNA, nomenclature (define ISR), and their different mechanisms of action (perhaps with a figure) than the cursory one in lines 119-128.
3. To some degree, a manuscript such as this will be a laundry list, but are there any patterns that can be deduced of sets of ncRNA that enhance or suppress the overall IFN responses?
3. Figure 1. GAF should only refer to STAT1 homodimers, correct? For accuracy, it is better to show mRNA transcribed from the genome rather than protein
4. Figure 2. This figure shows the effect of various nc RNA on downstream ISGs. It is comprehensive and difficult to follow. I suggest numbering the ncRNA in the legend and the figure so that the reader can find the ncRNA in the figure. Also while some ncRNA act on the protein, others regulate transcription or expression of the transcript. The figure doesn't distinguish the two mechanisms because all the ISGs are labeled with a pink star.
3. Lines 81-85 are a little unclear. If SOCS3 suppresses IFN and inflammatory responses (references 25 and 26), why would suppression of SOCS3 expression by STAT3 suppress proinflammatory gene expression?
4. Line 158: state how miR-155 affects SOCS1 activity--by interfering with the protein, decreasing transcripts?
5. Lines 160-167. It seems that the authors are suggesting that miR-221/222 may affect the balance of STAT1/STAT2 verses STAT3 homodimer signaling?
6. Lines 205-217; discuss importance of ISG15 before discussion of modulation of its expression would be clearer.
7. Line 219. The sentence isn't clear.
8. Again, the discussion can be shortened, and a lot of information, such as the concepts if IFN signaling and specificity of ncRNA to specific viruses or cells would be useful to understand up front rather than later in the manuscript.
9. The supplemental table seems to be missing some ncRNA discussed in the text. I don't understand the Deregulation column--this means that genes or expression are upregulated or downregulated by the ncRNA?
10. A table of ncRNA according to effect might be useful: i.e. positive or negative regulators of IFN expression, positive or negative regulators of downstream innate responses
Reviewer 2 Report
Min and colleagues write a review article on the role of non-coding RNAs (ncRNAs) in antiviral innate immune responses. It has been reported that upon viral infection and/or interferon (IFN) stimulation, many ncRNAs such as miRNAs and lncRNAs are expressed and regulate anti-viral innate immune responses via various mechanisms. This review is very interesting and comprehensive and well summarizes recent findings on how miRNAs, lncRNAs, and circRNAs are induced, regulate antiviral responses including IFN stimulated gene (ISG) expression and function in a positive and negative manner. There are some minor suggestions as listed below:
1. Supplemental Table may be included in the main body.
2. Figure 2: although the role of lncRNA CMPK2/NRIR is well explained in the legend, it’s a bit confusing visually in the figure. A legend (red arrow: promoting, blue: suppressive, green: either of these, context-dependent etc.) may be helpful.
3. The meaning “late-specific expression” (line 209) is unclear
4. Line 219, does “the latter” indicate probably ISGs? Please clarify.
5. The point of the paragraph (lines 225-235) on lncRNA CMPK2/NRIR may be a bit unclear. This lncRNA works as a negative regulator (hepatocytes and epithelial cells) and a positive regulator (monocytes) of IFN responses. Please clarify these points.
6. Please add citations to the paragraph 3.4 (RNA modification)
7. Conclusions mostly focus on circRNA. It would be helpful to have a broader summary, insights, and prospects for future research on ncRNA including miRNA and lncRNA.
8. May be helpful to explain abbreviation of viruses, HCV, IAV etc.
9. Some typos: TRL (lines 65, 66). Line 189, INF. ISG15 (line 207) may be lncISG15. Line 240, miRNA mRNA. Line 324, Interesting(ly).
Round 2
Reviewer 1 Report
Please check supplementary table #1. the title of one of the columns is "Deregulation" although you stated in comment to reviewer that you changed it to "Regulation."
